# Overexpression of a *Malus baccata* MYB Transcription Factor Gene *MbMYB4* Increases Cold and Drought Tolerance in *Arabidopsis thaliana*

**DOI:** 10.3390/ijms23031794

**Published:** 2022-02-04

**Authors:** Chunya Yao, Xingguo Li, Yingmei Li, Guohui Yang, Wanda Liu, Bangtao Shao, Jiliang Zhong, Pengfei Huang, Deguo Han

**Affiliations:** 1Key Laboratory of Biology and Genetic Improvement of Horticultural Crops (Northeast Region), Ministry of Agriculture and Rural Affairs, National-Local Joint Engineering Research Center for Development and Utilization of Small Fruits in Cold Regions, College of Horticulture & Landscape Architecture, Northeast Agricultural University, Harbin 150030, China; 13753449701@163.com (C.Y.); lxguo@126.com (X.L.); Mei990304@163.com (Y.L.); shaobangtao@163.com (B.S.); liang096633@163.com (J.Z.); hg3194044267@163.com (P.H.); 2Horticulture Branch of Heilongjiang Academy of Agricultural Sciences, Harbin 150040, China; haaslwd@126.com

**Keywords:** *Malus baccata* (L.) Borkh, *MbMYB4*, cold stress, drought stress

## Abstract

In the natural environment, plants often face unfavorable factors such as drought, cold, and freezing, which affect their growth and yield. The MYB (v-myb avian myeloblastosis viral oncogene homolog) transcription factor family is widely involved in plant responses to biotic and abiotic stresses. In this study, *Malus baccata* (L.) Borkh was used as the research material, and a gene *MbMYB4* of the MYB family was cloned from it. The open reading frame (ORF) of *MbMYB4* was found to be 762 bp, encoding 253 amino acids; sequence alignment results and predictions of the protein structure indicated that the MbMYB4 protein contained the conserved MYB domain. Subcellular localization showed that MbMYB4 was localized in the nucleus. In addition, the use of quantitative real-time PCR (qPCR) technology found that the expression of *MbMYB4* was enriched in the young leaf and root, and it was highly affected by cold and drought treatments in *M. baccata* seedlings. When *MbMYB4* was introduced into *Arabidopsis thaliana*, it greatly increased the cold and drought tolerance in the transgenic plant. Under cold and drought stresses, the proline and chlorophyll content, and peroxidase (POD) and catalase (CAT) activities of transgenic *A. thaliana* increased significantly, and the content of malondialdehyde (MDA) and the relative conductivity decreased significantly, indicating that the plasma membrane damage of transgenic *A. thaliana* was lesser. Therefore, the overexpression of the *MbMYB4* gene in *A. thaliana* can enhance the tolerance of transgenic plants to cold and drought stresses.

## 1. Introduction

In nature, the growth and development of plants are often affected by the environment, where the harsh natural environment may even cause irreversible damage to plants, leading to plants’ death [1]. In recent years, the development of molecular biology has allowed people to deeply understand the resistance mechanism of plants to adversity stress in terms of gene expression, transcriptional regulation, and signal transduction [2]. Among them, transcriptional regulation plays a connecting role in plant stress response. Transcriptional regulation means that transcription factors regulate the expression of a series of genes by binding to the cis-acting elements in the promoter region, and it is an important part of the plant stress response mechanism [3,4].

The MYB (v-myb avian myeloblastosis viral oncogene homolog) transcription factor family is one of the largest transcription families in plants [5,6], and it plays an important role in plant growth, development, and metabolic regulation [7]. The R2R3-MYB transcription factor is the most abundant MYB protein in plants. Its DNA binding region is composed of two homologous MYB domains (R2, R3) and binds to the target sequence through the synergistic effect of R2 and R3 specifically [8,9]. Most R2R3-MYB proteins have transcriptional activation domains at the C-terminus, which play a key role in the process of plant cell differentiation, hormone response, secondary metabolism, and resistance to biotic and abiotic stresses [10,11,12].

Cold is a common abiotic stress during plant growth and development. Cold stress will increase the permeability of plant cell membranes, leading to the extravasation of solutes in the membrane, imbalance of water metabolism, and abnormal photosynthesis and respiration rates [13]. It further causes plant leaves to wither, which seriously affects plant growth and yield [14,15]. MYB transcription factor genes can play a key role in plant response to cold stress. Studies have shown that *OsMYB2* encodes a type of R2R3 transcription factor in rice, and the overexpression of this gene in rice can enhance the resistance of transgenic plants to cold [16]; *MdMYB108L* can regulate the expression of downstream CBF genes, thereby improving the cold resistance of *Malus domestica* [17]; *MdMYB23* can enhance the cold resistance of *M. domestica* mainly by the accumulation of proanthocyanidins [18], and similar findings have also been found in *Arabidopsis thaliana* and maize [19,20].

During the growth of plants, drought and salt stress usually occur concurrently. At present, many MYB genes have been isolated and cloned in *A. thaliana*, tobacco, and some food and commercial crops [5,21], where the salt and drought resistance of transgenic plants are significantly improved due to the expression of these genes. Studies have shown that the expression of *AtMYB20* is induced by high salt and drought stress; under stress conditions, the chlorophyll content of *AtMYB20* transgenic plants is significantly higher than that of wild-type (WT); and the water loss rate is lower than that of WT, indicating that overexpression of *AtMYB20* can improve the salt and drought tolerance of transgenic plants [22]. Overexpression of the *OsMYB48-1* gene in rice increased the expression of abscisic acid (ABA) synthesis genes (*OsNCED4*, *OsNCED5*) in transgenic plants, and the survival rate of transgenic plants was significantly higher than that of WT, indicating that the *OsMYB48-1* gene regulates ABA accumulation to improve the drought resistance of transgenic plants [23].

## 2. Results

### 2.1. Cloning and Bioinformatics Analysis of MbMYB4

ExPASy ProtParam analysis results showed that the full length of the *MbMYB4* gene was 762 bp and encoded 253 amino acids (Appendix A). The predicted theoretical molecular weight (MW) of the protein was 28.843 kDa, the theoretical isoelectric point (pI) was 6.84, and the average hydrophilic coefficient was −0.839, which indicated that the protein was hydrophilic. Among the amino acids contained in the MbMYB4 protein, Ser (12.0%), Asn (8.0%), Lys (7.2%), and Glu (6.8%) accounted for a large proportion.

MYB amino acid sequences of other species were compared with the MbMYB4 protein sequence. The results showed that the MbMYB4 protein and other MYB proteins had roughly the same R2 and R3 conserved domains (Figure 1A). Furthermore, the phylogenetic tree showed that the MbMYB4 protein had the highest homology with MdMYB4 (*M. domestica*) (Figure 1B).

As shown in Figure 2A, the secondary structure of the MbMYB4 protein contained 25.90% of α helix, 4.38% of β turn, 66.14% of random coils, and 3.59% of extended chain. The amino acid sequence of MbMYB4 had two SANT conserved domains (Figure 2B), and it was speculated that MbMYB4 belongs to the R2R3-MYB family. Using the SWISS-MODEL online analysis tool to predict the tertiary structure of MbMYB4, it was found that the protein had a HTH structure, which was consistent with the predicted result of its secondary structure (Figure 2C).

### 2.2. Subcellular Localization of MbMYB4 Protein

Based on the subcellular localization, the specific location of MbMYB4 protein functioning in the cell can be determined. Under the fluorescence confocal microscope (LSM 900, Precise, Beijing, China), the green fluorescent protein (GFP) as a control was distributed throughout the cell (Figure 3B), while the fluorescence of the *MbMYB4*-GFP fusion protein was only distributed in the nucleus (Figure 3E). In addition, observing the condition of the cell nucleus after DAPI staining can further prove that the MbMYB4 protein was located in the cell nucleus (Figure 3C,F).

### 2.3. Expression Analysis of MbMYB4 in Malus baccata

Analyzing the expression pattern of the *MbMYB4* gene found that *MbMYB4* can be expressed in the young leaf, stem, root, and mature leaf, but the expression levels in these parts were different. Under control conditions, the expression of *MbMYB4* was higher in the root and young leaf than the mature leaf and stem of *M. baccata* seedlings (Figure 4A). Under the five stresses of low-temperature, high-salt, dehydration, high-temperature, and ABA, the expression level of *MbMYB4* in the young leaf increased first and then decreased within 12 h after treatment, and the expression level peaked at 3 h, 7 h, 7 h, 3 h, and 5 h of treatment, respectively (Figure 4B). Similarly, the expression level in the root also showed the same trend, reaching peaks at 5 h, 5 h, 3 h, 5 h, and 3 h of treatment, respectively (Figure 4C), and under cold and drought stresses, the expression levels of *MbMYB4* in the young leaf and root were higher than those under other stress treatments.

### 2.4. Overexpression of MbMYB4 in A. thaliana Improved Cold Tolerance

To investigate the role of *MbMYB4* in response to cold stress, *MbMYB4*-overexpressing transgenic *A. thaliana* was generated. Using wild-type (WT) and empty vector (UL) lines as controls, quantitative real-time PCR (qPCR) analysis was performed on six T_2_ transformed lines (S1, S2, S3, S4, S5, and S6 in Figure 5A). The results showed that the target fragment did not appear in WT and UL, while the six T_2_ transgenic plants amplified bands consistent with the target gene fragment, indicating that the target gene had been integrated into T_2_ generation transgenic lines (S1, S2, S3, S4, S5, and S6 in Figure 5A). As, among the six transgenic lines, S1, S3, and S5 have higher DNA relative expression levels, these three lines were cultured to the T_3_ generation of transgenic *A. thaliana*.

WT, UL, and T_3_ transgenic *A. thaliana* were cultured in the light incubator for about 3 weeks, moved to −6 °C for cold treatment for 14 h, and finally cultivated under normal conditions for 7 days. The results showed that cold treatment caused different levels of damage to transgenic *A. thaliana* (S1, S3, and S5), UL, and WT (Figure 5B), but WT and UL *A. thaliana* were more severely damaged, and their leaves wilted more deeply. Seven days after the stress was removed, most of WT and UL died, the survival rates were only 36% and 26%, while the average survival rate of transgenic *A. thaliana* was 77% (Figure 5C).

In addition, we measured the content of MDA, chlorophyll, and proline, the POD and CAT activity, and relative conductivity in WT, UL, and transgenic *A. thaliana* (S1, S3, and S5) under normal conditions (25 °C) and cold treatments (Figure 6). There was almost no difference in the physiological indicators of each plant before treatment. After cold treatment, the activity of POD and CAT and the content of proline increased in all plants, but in contrast, the increases in these indexes were greater in transgenic *A. thaliana*; the content of MDA and relative conductivity also increased due to cold stress, while the increase in transgenic *A. thaliana* was lower than those in WT and UL; the chlorophyll content of all *A. thaliana* lines decreased under cold conditions, but the chlorophyll content of transgenic *A. thaliana* decreased slighter compared with WT and UL. These indicate that the expression of *MbMYB4* improves the cold resistance of transgenic *A. thaliana*.

### 2.5. Expression Analysis of Cold Tolerance-Related Genes in A. thaliana Overexpressing MbMYB4

The molecular regulation pathway of *A. thaliana* responding to cold stress mainly relies on the CBF (CRT/DRE-binding factor) transcription factor [24]. Therefore, we analyzed the expression changes of several key genes *AtCBF1*, *AtCBF3*, *AtRD29a,* and *AtCOR15a* located downstream of the MYB transcription factor under cold treatment (Figure 7). After 14 h of cold treatment at −6 °C, the expression levels of *AtCBF1*, *AtCBF3*, *AtRD29a,* and *AtCOR15a* in *MbMYB4*-overexpression transgenic lines were significantly higher than those of WT and UL lines, indicating that the *MbMYB4* transcription factor positively regulates *AtCBF1* and *AtCBF3*. Then, we further promoted the expression of *AtRD29a* and *AtCOR15a* to improve the resistance of plants to cold stress.

### 2.6. Overexpression of MbMYB4 in A. thaliana Improved Drought Tolerance

In order to understand the role of *MbMYB4* in response to drought stress, we stopped watering transgenic *A. thaliana* (S1, S3, and S5 in Figure 8A), UL, and WT for 10 days to simulate drought stress. Under normal growth conditions, the transgenic lines have the same growth state as WT and UL plants. After stopping watering for 10 days, the leaves of all plants turned slightly yellow. Then, after normal watering (stress removed) for 7 days, the state of transgenic lines was significantly better than that of other lines (Figure 8A). The survival rates of S1, S3, and S5 reached 76%, 74%, and 85%, respectively, while WT and UL survival rates were only 27% and 49%, respectively (Figure 8B).

In addition, the measurement of the physiological indicators of all *A. thaliana* under normal and drought conditions found that when plants were under sufficient water conditions, there were no significant differences in the physiological indexes of WT, UL, and transgenic *A. thaliana*, but when plants were under drought stress, the activity of POD, CAT, and the content of proline in transgenic *A. thaliana* were higher than those in WT and UL plants. The increase in MDA content and relative conductivity and the decrease in chlorophyll content in WT and UL *A. thaliana* were greater than those of transgenic *A. thaliana* (Figure 9). The results showed that the expression of the *MbMYB4* gene increased the resistance of transgenic *A. thaliana* to drought stress.

### 2.7. Expression Analysis of Drought Tolerance-Related Genes in A. thaliana Overexpressing MbMYB4

The results of the analysis of *MbMYB4* expression showed that ABA can induce the expression of *MbMYB4.* The expression of the ABA synthesis gene *AtNCED3* and the ABA signal transduction-related gene *AtSnRK2.4* in *MbMYB4* transgenic *A. thaliana* was further explored (Figure 10), and it was found that under normal conditions, the expression levels of *AtNCED3* and *AtSnRK2.4* were not significantly different among all *A. thaliana* lines, but under drought stress, the expression levels of *AtNCED3* and *AtSnRK2.4* in *MbMYB4*-overexpression lines were significantly higher than those of WT and UL lines, indicating that *MbMYB4* can participate in the response of plants to drought stress through two pathways: regulating ABA synthesis and ABA signal transduction. In addition, the expression of *AtCAT1* and *AtP5CS* genes increased significantly under drought stress. This indicates that *MbMYB4* can regulate the key genes of drought stress to improve the scavenging ability of reactive oxygen species (ROS), thereby enhancing the drought resistance of plants.

## 3. Discussion

MYB transcription factors in plants contain a conserved MYB domain, and its DNA binding domain contains 1 to 4 incompletely repeated amino acid sequence structures, called R structures (R1, R2, and R3). The R structure is a folded protein composed of about 50 amino acids, including a series of highly conserved amino acid residues and spacer sequences, among which the amino acid residues participate in the DNA binding process in the form of helix-turn-helix (HTH) [25,26,27]. The R2R3-MYB protein is the largest MYB protein subclass in plants, and it contains two conserved domains, R2 and R3. This experiment used *M. baccata* as the test material, and its genome has not been sequenced. Therefore, using *MdMYB25* (NM_001293983.1, *M. domestica*) as the reference sequence, specific primers were designed to amplify the target gene *MbMYB4*. Sequence analysis revealed that the ORF of *MbMYB4* is 762 bp, encoding 253 amino acids; the average hydrophilic coefficient of the protein is -0.839, which is a hydrophilic protein (Appendix A). Analysis of the conserved domains of MbMYB4 protein revealed that there are two SANT-MYB DNA binding domains, which is speculated to belong to the R2R3-MYB family (Figure 1A). Compared with the highly conserved MYB domain, other regions of the R2R3-MYB protein have a high degree of variability [28]. The MbMYB4 protein sequence was compared with the MYB protein sequences of other species. The results showed that the MbMYB4 protein is highly similar to the conserved sequences of other MYB family proteins, but the nonconserved region is quite different. This feature is consistent with characteristics of transcription factors. The evolutionary tree showed that MbMYB4 is most related to *M. domestica* MdMYB4 (Figure 1B).

Studies have shown that the functional area of transcription factors in plants is mainly located in the nucleus. Han et al. found that MbNAC transcription factor proteins are located in the nucleus [29,30]. After constructing the *MbMYB4*-GFP fusion transient expression vector, it was introduced into onion epidermal cells by the particle bombardment method, the distribution of the tag protein in the cells was observed under a confocal microscope, and the results showed that the MbMYB4 protein was localized in the nucleus (Figure 3). This is also consistent with the distribution of the MYB protein in the *M. domestica* by previous studies [31,32].

MYB transcription factors are involved in the growth and development of plants. For example, *AtMYB33* and *AtMYB65* are expressed differently in various tissues and organs [33]; *AtMYB118* is highly expressed in the leaf and participates in leaf morphogenesis [34]. In this study, the expression levels of *MbMYB4* in the young leaf and root of *M. baccata* were higher (Figure 4A), indicating that it may play an important role in the growth and development of the root and young leaf. The results of this study showed that abiotic stresses such as low-temperature, high-salt, dehydration, high-temperature, and ABA all induce the expression of *MbMYB4*, and the expression of the gene under different stresses changes with the treatment time. Under low-temperature stress and dehydration stress, the expressions of *MbMYB4* in the young leaf and root were upregulated most significantly (Figure 4B,C). Under low-temperature stress, the expression levels of *MbMYB4* in the young leaf and root reached their peaks at 3 h and 5 h of treatment, and they were 7.8 times and 7.9 times higher than those in untreated groups, respectively. Under dehydration stress, the expression levels of *MbMYB4* in the young leaf and root reached their peaks at 7 h and 3 h of treatment, and the expression level was 8.7 times and 7.5 times those of the untreated group, respectively. Most MYB transcription factors involved in the response to drought stress require ABA as a medium [35]. Under ABA stress, the expression levels of *MbMYB4* in the young leaf and root were significantly different from the control at 5 h and 3 h, and the expression levels were significantly higher than those of the untreated group. In high-salt and high-temperature stress, *MbMYB4* in the root was more sensitive to high-salt stress (Figure 4C). This showed that when *M. baccata* are in a growth environment with a high salt concentration, the root receives the salt stress signal first. Therefore, the root can give priority to responding to adverse conditions.

After transforming the model plant *A. thaliana* with *MbMYB4*, WT, UL, and transgenic lines were treated with cold and drought. After cold and drought stresses, all plants showed damage. However, the degree of yellowing and wilting of transgenic *A. thaliana* was lighter. Statistics of their survival rate showed that the survival rate of transgenic *A. thaliana* is higher than those of WT and UL *A. thaliana*. The results showed that *MbMYB4* improves the cold and drought tolerance of transgenic plants.

In order to explore the mechanism by which this gene enhances plant cold and drought tolerance, it is necessary to study its physiological, biochemical, and molecular regulation levels. When plants are under cold stress, their cell plasma membranes become more permeable, electrolytes and soluble substances leak out, and electrical conductivity increases. Therefore, the measurement of relative electrical conductivity is a method used to determine the resistance of plants to stress [36]. Under cold and drought stresses, compared with WT and UL, the relative conductivity of overexpression transgenic lines has a smaller increase, indicating that the cell membrane has less damage and plants have less damage from stress. According to the phenotypic analysis of plants under cold and drought stresses, cold stress can decompose the chlorophyll in plants and turn leaves yellow, make plants unable to photosynthesize, and die in severe cases. Therefore, the chlorophyll content in plants can also be used as a measure of plant resistance [37]. Under stress conditions, the chlorophyll content of transgenic plants was indeed significantly higher than that of wild type. In addition, plants subjected to cold and drought stresses can also enhance the water holding capacity by accumulating osmotic adjustment substances [38]. Proline is an important type of osmotic adjustment substance. Proline in plants can promote protein hydration and increase the content of soluble protein [39], thereby maintaining enzyme activity under cold and drought conditions. According to the changes in the proline content of each line, it can be seen that the cytoplasm of transgenic plants suffers less damage under cold and drought conditions. Due to the high level of ROS in the body when plants are dehydrated, which causes membrane damage, protective enzymes such as SOD, POD, and CAT begin to work to remove excessive reactive oxygen species to maintain intracellular balance. At the same time, studies have found that the activities of POD and CAT in the root of cold-acclimated rice are significantly enhanced, indicating that these antioxidant enzymes also play the most important role in cold resistance [40]. The results of the study showed that, compared with WT and UL A. thaliana, the activity of CAT and POD in transgenic *A. thaliana* with *MbMYB4* was significantly increased, and the content of MDA was significantly reduced. This shows that transgenic *A. thaliana* has a stronger ability to scavenge superoxide ions and a stronger tolerance to cold and drought.

Studies have shown that MYB transcription factors can bind to the CBF promoter region [41,42]. CBF induces the expression of downstream genes such as CORs, RDs, and LTIs by combining CRT/DRE cis-acting elements to improve the cold resistance of plants [43,44,45,46,47,48]. The expressions of two key genes *CBF1* and *CBF3* [44] in the CBF-dependent pathway and their downstream cold-responsive genes *COR15a* [14] and *RD29a* [49] were analyzed under normal conditions and cold treatment. The results showed that *MbMYB4* can promote the expression of *AtCBF1*, *AtCBF3*, *AtCOR15a*, and *AtRD29a* in transgenic *A. thaliana*. Therefore, *MbMYB4* can participate in the response to cold stress through a CBF-dependent pathway.

The ABA signal transduction pathway is an important pathway in plant drought stress response. Under drought stress, plants can induce stomatal closure by synthesizing ABA, thereby reducing water loss and improving drought tolerance. In *A. thaliana*, a heterologous expression of *MYB10* has been shown to increase ABA hypersensitivity and enhance drought tolerance [22,50]. The expressions of ABA synthesis-related gene *AtNCED3* and ABA signal transduction-related gene *AtSnRK2.4* in transgenic *A. thaliana* were analyzed, and it was found that their expressions were upregulated under drought conditions. This indicated that *MbMYB4* could participate in the process of response to drought by regulating ABA synthesis and ABA signal transduction. In addition, the overexpression of *GaMYB85* can promote the accumulation of free proline in transgenic *A. thaliana*, and the drought stress-responsive genes *RD22*, *ADH1*, *RD29A*, *P5CS*, and *ABI5* are also induced to express, thereby enhancing the resistance of plants to drought and salt stress [51]. Drought stress can induce the expression of *CAT1* and significantly increase the activity of catalase. *CAT1* acts as a major scavenger of H_2_O_2_ and plays an important role in adaptation to drought stress [52]. The expression of *MbMYB4* in transgenic *A. thaliana* induces the expression of the drought stress-responsive genes *AtCAT1* and *AtP5CS*, thereby enhancing the resistance of plants to drought stress.

In summary, we can derive a potential model to describe the role of *MbMYB4* in cold and drought stresses based on the above results and previous studies (Figure 11). First, cold stress induces the expression of *MbMYB4* so that it can bind to the CBF promoter region. Two key genes in the CBF-dependent pathway, *CBF1* and *CBF3*, directly activate their expression by binding to the CRT/DRE cis-acting elements of the downstream cold-responsive genes *COR15a* and *RD29a*, enhancing the cold resistance of plants. Secondly, drought stress induces the expression of *MbMYB4*, regulates the up-regulated expression of *NCED3* and *SnRK2.4*, and promotes ABA biosynthesis and signal transduction, thereby enhancing the drought resistance of transgenic plants. At the same time, the production of ABA can also induce the expression of *MbMYB4*. In addition, under stress conditions, the expressions of drought-responsive key genes *CAT1* and *P5CS* increase, indicating that *MbMYB4* can regulate key genes under drought stress to reduce the accumulation of reactive oxygen species, and can also increase the level of proline to improve the water-holding capacity of plant cells under drought conditions, thereby enhancing the drought resistance of transgenic plants.

## 4. Materials and Methods

### 4.1. Plant Material and Growth Conditions

The test-tube plantlets of *M. baccata* (Appendix A) were propagated on MS growth medium (MS + 0.6 mg/L 6-Benzylaminopurine (6-BA) + 0.6 mg/L Indole-Butytric acid (IBA)) for 30 days, and then transferred to MS rooting medium (MS + 1.2 mg/L IBA) to continue culturing until white roots grew. Finally, the seedlings were transferred to Hoagland solution for 40 days for growth. The solution was changed every 2–3 days. The temperature of the culture chamber was maintained at 20 °C and the relative humidity was maintained at about 85%. When the seedlings had 7–9 fully expanded leaves and the roots were well developed, we selected 60 seedlings with better morphology and basically the same growth. First, we sampled the four different parts of the young leaf, root, mature leaf, and stem of 10 seedlings, then the remaining 50 seedlings were divided into five groups for different stress treatments: low-temperature stress (hydroponic seedlings (Appendix A) were cultured at 4 °C), salt stress (hydroponic seedlings were cultured in Hoagland solution containing 200 mM of NaCl), dehydration stress (hydroponic seedlings were cultured in Hoagland solution with a concentration of 20% PEG6000), high-temperature stress (hydroponic seedlings were cultured in a lighted incubator at 37 °C), and ABA stress (hydroponic seedlings were cultured in Hoagland solution with ABA concentration of 50 μM). After 0, 1, 3, 5, 7, 9, and 12 h of stress treatment, we took samples of the young leaf and root of these seedlings, and the obtained plant materials were frozen with liquid nitrogen and stored at −80 °C for RNA extraction.

### 4.2. Isolation and Cloning of MbMYB4

The OminiPlant RNA Kit (Kangweishiji, Beijing, China) was used for RNA extraction. Then, RNA was used as the template to synthesize the first-strand cDNA with TransScript^®^ First-Strand cDNA Synthesis SuperMix (TransGen Biotech, Beijing, China) [53]. RNA and cDNA were assessed by 1.0% agarose gel electrophoresis [54]. The CDS region of *MdMYB25* (NM_001293983.1, *M. domestica*) was used as the reference sequence, and primers (*MbMYB4*-F and *MbMYB4*-R, Appendix A) were designed with Primer 5.0 software. After the primers were synthesized, they were used for the amplification of *MbMYB4*. The purified DNA was linked to the pEASY^®^-T1 Cloning Kit (TransGen Biotech, Beijing, China) and sequenced [55,56].

### 4.3. Bioinformatics Analysis of MbMYB4

DNAMAN5.2 was used to translate the *MbMYB4* gene sequence, the amino acid sequence of the MbMYB4 protein was compared with MYB sequences with higher homology in other species, and ExPASy (https://web.expasy.org/protparam/, accessed on 31 January 2022) was used to predict the primary structure of the MbMYB4 protein. The domain of the MbMYB4 protein was predicted on the SMART website (http://smart.embl-heidelberg.de/, accessed on 31 January 2022), and the SWISS-MODEL (https://swissmodel.expasy.org/, accessed on 31 January 2022) was used to predict the tertiary structure of the MbMYB4 protein. MEGA7.0 (http://www.megasoftware.net, accessed on 5 April 2021) was used to construct a homologous phylogenetic tree of MbMYB4.

### 4.4. Subcellular Localization Analysis of MbMYB4 Protein

The upstream and downstream primers with *BamH*I and *Sal*I restriction sites were designed (*site*-F and *site*-R, Appendix A), and the DNA fragments of *MbMYB4* were cloned into *BamH*I and *Sal*I sites of the pSAT6-GFP-N1 vector. The recombinant plasmid containing the *MbMYB4* gene was injected into onion epidermal cells by particle bombardment [57]. After overnight culture, it was placed under a confocal microscope to observe the fluorescent signals of the control protein and the *MbMYB4*-GFP fusion protein, to observe the location of the protein encoded by the gene in the cells. DAPI staining was used as a nucleus marker for nucleus detection.

### 4.5. Quantitative Real-Time PCR (qPCR) Analysis of MbMYB4

The expressions of the *MbMYB4* gene under different treatments were determined by qPCR with a pair of specific primers (*MbMYB4*-qF and *MbMYB4*-qR, Appendix A). The PCR condition was as follows: 30 s at 94 °C; 40 cycles of 5 s at 95 °C, 40 s at 54 °C, and 30 s at 72 °C; and then 10 min at 72 °C. The expression of the *MbMYB4* gene was detected by the TB Green^™^ Premix Ex Taq^™^ II (Tli RNaseH Plus kit) (TaKaRa, Beijing, China) according to the manufacturer’s protocol, and the *Actin* gene (NC_024251.1, *M. domestica*) was used as the internal reference gene. *MbMYB4* expression level was analyzed using the 2^−ΔΔCT^ method.

### 4.6. Vector Construction and Agrobacterium-Mediated A. thaliana Transformation

According to the principle of homologous recombination to connect the expression vector, the primers including the target fragment-specific primer sequence, restriction sequence (*BamH*I and *Sal*I), and overlapping sequence were designed (*HR*-F and *HR*-R, Appendix A). The target fragment was ligated with the digested pCAMBIA2300 plasmid and transformed into *Agrobacterium tumefaciens* GV3101 [58].

The right amount of Colombian WT *A. thaliana* seeds was prepared and fully disinfected with 75% alcohol and 10% NaClO and then dried. The seeds were sown evenly on the sowing medium (1/2MS + 20% sucrose + 0.7% agar) until 2–4 leaves grew. We selected seedlings with well-developed roots and transferred them to nutrient bowls (ratio of turfy soil to vermiculite is 2:1), and then cultivated them in a light incubator (temperature setting: 25 °C daily/22 °C night, humidity 60%, 16 h light/8 h darkness). After WT *A. thaliana* entered the blooming period, we infected it by the inflorescence dip method and harvested the seeds [59]. The harvested seeds were evenly sown on the screening medium (4.3 g/L MS + 30 g/L sucrose + 7.5 g/L agar + 50 mg/L kanamycin) [60] until T_3_ generation. We continued to sow T_3_ generation seeds on MS medium containing antibiotics (50 mg/L kanamycin). After a period of time, the selection medium was full of positive seedlings. At this time, the transgenic *A. thaliana* obtained was the homozygous transgenic line. Transgenic *A. thaliana*, UL, and WT were transplanted into nutrient bowls and grown in a light incubator.

### 4.7. Determination of Related Physiological Indicators

WT, UL, and transgenic *A. thaliana* were divided into two groups after three weeks of growth. One group was treated at −6 °C for 14 h to cause cold stress; the other group was not watered for 10 d, thereby causing drought stress to them. After the stress treatment, the *A. thaliana* of each group was cultivated under normal conditions for one week, and the morphological changes of each group under stress and normal environments were observed. Fresh leaves were immersed in an ethanol-acetone mixture solution for 24 h and the chlorophyll content was calculated by measuring the absorbance value at 645 and 663 nm of the mixed solution using a spectrophotometer [61]. Proline was extracted with sulfosalicylic acid, and the content was calculated using the method described by Huang et al. [62]. The *A. thaliana* leaves were put into deionized water and were repeatedly pumped and deflated until the leaves were completely submerged in deionized water. We measured the conductance of the extract solution with a conductivity meter, heated it in a boiling water bath for 30 min, cooled it down sufficiently, measured the conductance again, and used the formula to calculate the relative conductivity [63]. The determination of POD activity was based on the guaiacol method [64]. CAT activity was measured following the ultraviolet (UV) absorption method [65]. MDA activity was measured by the colorimetric method using a spectrophotometer [66]. Calculation formulas for all indicators are listed in Appendix A.

### 4.8. Analysis of Downstream Gene Expression of MbMYB4

The mRNAs of WT, UL, and *MbMYB4* transgenic *A. thaliana* under normal conditions and after cold and drought stresses were extracted and reverse-transcribed into the first-strand cDNA as the template, and *AtActin* was used as an internal reference. Specific primers were designed (Appendix A) to perform qPCR experiments on several important regulatory genes downstream of MYB transcription factors: cold-responsive key genes (*AtCBF1*, *AtCBF3*, *AtCOR15a*, *AtRD29a*) and drought-responsive related genes (*AtNCED3*, *AtSnRK2. 4*, *AtCAT1*, *AtP5CS*).

### 4.9. Statistical Analysis

SPSS software was used to analyze the differences with Duncan’s multiple range tests. Statistical differences were referred to as significant when **p* ≤ 0.05, ** *p* ≤ 0.01.

## Figures and Tables

**Figure 1 ijms-23-01794-f001:**
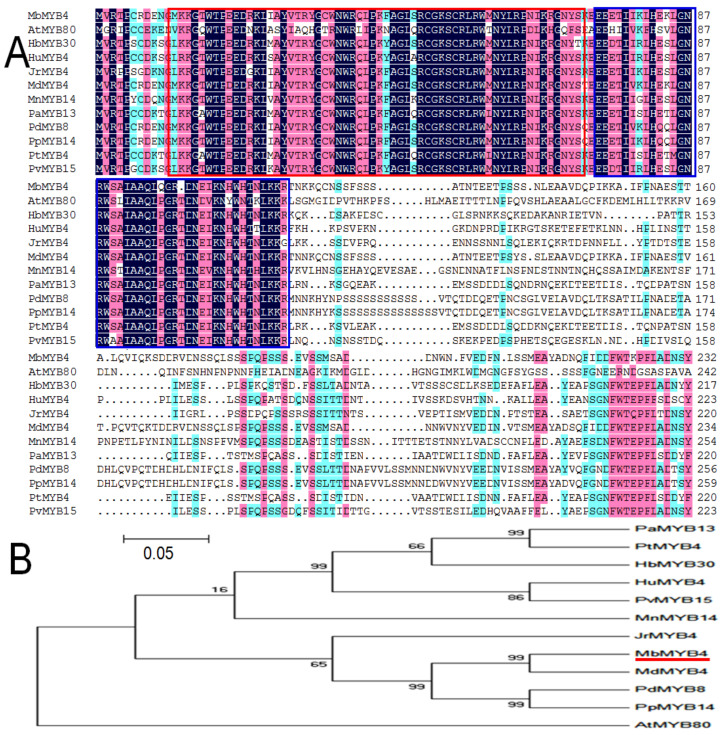
Comparison and phylogenetic relationship of MbMYB4 with MYB transcription factors in other species. (**A**) Homologous comparison of MbMYB4 protein with MYB proteins from other plant species. The sequence in the red and blue frame is the conserved amino acid sequence. (**B**) Phylogenetic tree analysis of MbMYB4 (marked by the red line) and other plant MYB proteins. The accession numbers are as follows: MdMYB4 (*Malus domestica*, NM_001293983.1), PdMYB8 (*Prunus dulcis*, XP_034208458.1), PpMYB14 (*Prunus persica*, XP_007216710.1), JrMYB4 (*Juglans regia*, XP_018821927.1), PaMYB13 (*Populus alba*, XP_034887613.1), PtMYB4 (*Populus trichocarpa*, XP_002319935.1), HbMYB30 (*Hevea brasiliensis*, XP_021689975.1), HuMYB4 (*Herrania umbratica*, XP_021296898.1), PvMYB15 (*Pistacia vera*, XP_031282890.1), MnMYB14 (*Morus notabilis*, XP_010095861.1), and AtMYB80 (*Arabidopsis thaliana*, NC_003076.8).

**Figure 2 ijms-23-01794-f002:**
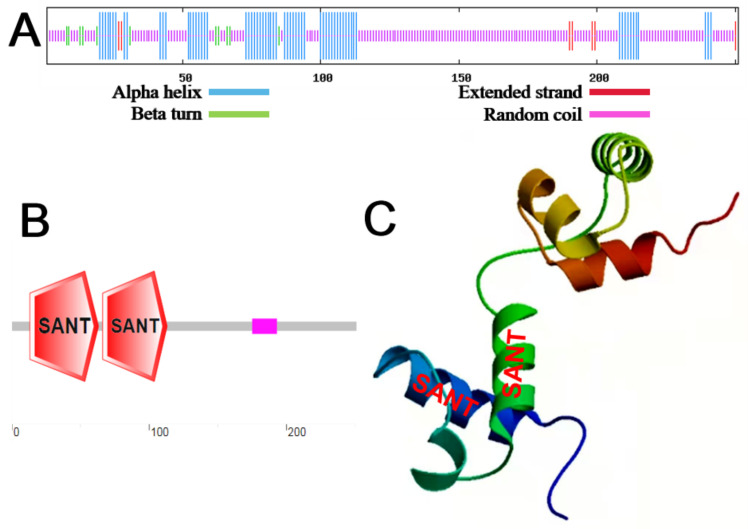
Secondary structure, domains, and tertiary structure prediction of MbMYB4 protein. (**A**) Prediction of the secondary structure of MbMYB4 protein. (**B**) Prediction of the domain of MbMYB4 protein. (**C**) Prediction of tertiary structure of MbMYB4 protein, where the locations of the red words in the figure are the possible positions of two SANT conserved domains of (**B**).

**Figure 3 ijms-23-01794-f003:**
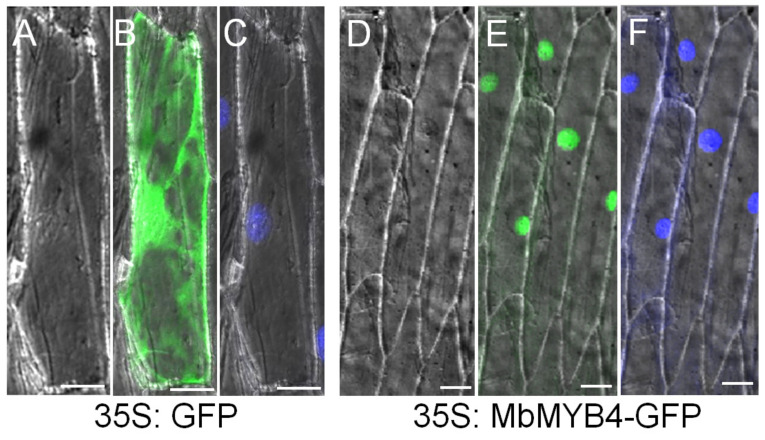
Subcellular localization of MbMYB4 protein. The 35S-GFP and 35S-*MbMYB4*-GFP translational products were expressed in onion epidermal cells and visualized by fluorescence microscopy in bright light (**A**, **D**), in dark field for GFP (**B**, **E**), and DAPI staining images (**C**, **F**). Scale bar corresponds to 5 μm. The experiment was repeated three times.

**Figure 4 ijms-23-01794-f004:**
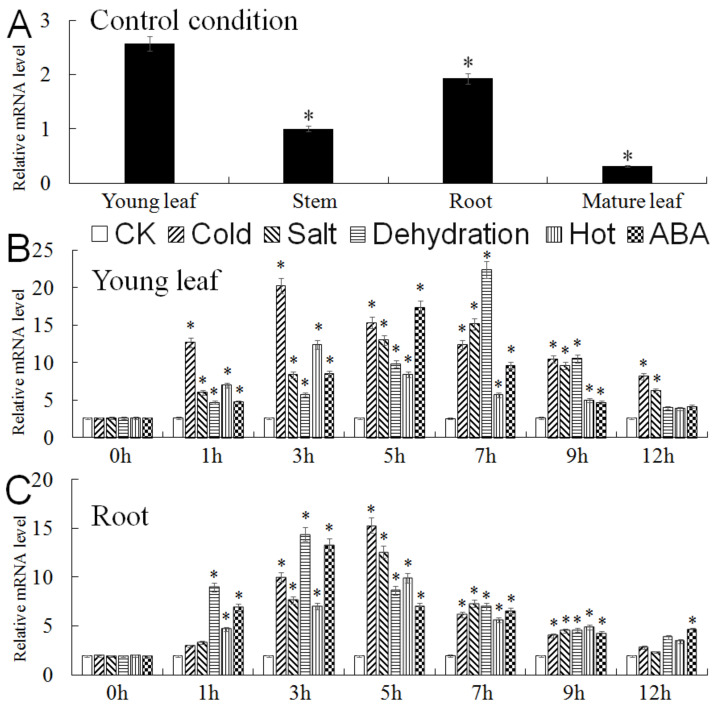
Expression of the *MbMYB4* gene in different organs of *Malus baccata*. (**A**) Expression of the *MbMYB4* gene in different organs under control condition. The expression level of young leaf was used as control. (**B**) Expression of the *MbMYB4* gene in the young leaf and (**C**) in the root under control condition (CK), low-temperature, high-salt, dehydration, high-temperature, and ABA. Data represent means of three replicates. The error bars represent standard deviation. Asterisks above the error bars indicate a significant difference between the treatment and control (0 h) (* *p* ≤ 0.05).

**Figure 5 ijms-23-01794-f005:**
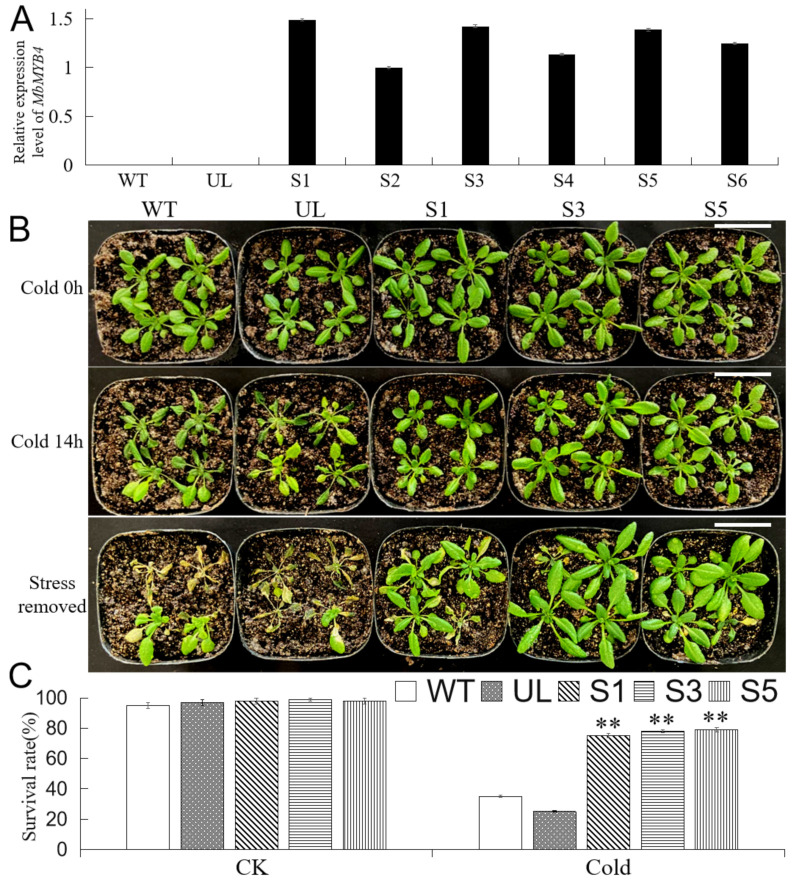
Overexpression of *MbMYB4* in *A.*
*thaliana* improved cold tolerance. (**A**) Relative expression level of *MbMYB4* gene in all *A.*
*thaliana* lines. (**B**) Phenotypes of wild-type (WT), empty vector (UL), and *MbMYB4* transgenic *A. thaliana* lines (S1, S3, and S5) under control condition, cold stress, and stress-removed condition. Scale bar corresponds to 4 cm. (**C**) Survival rates of WT, UL, and transgenic lines after control condition (CK) and cold stress. Data represent means of three replicates. Asterisks above columns indicate significant difference compared to WT (** *p* ≤ 0.01).

**Figure 6 ijms-23-01794-f006:**
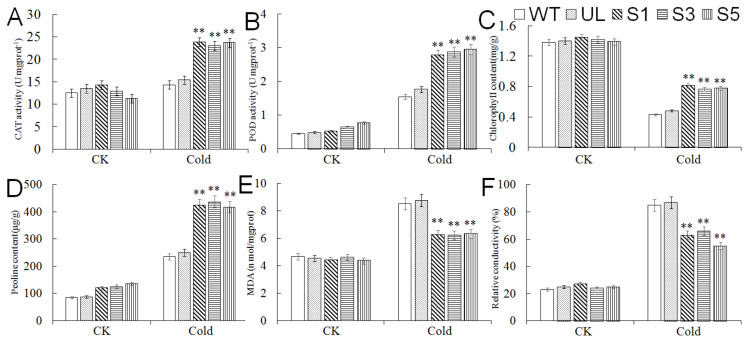
The levels of (**A**) CAT activity, (**B**) POD activity, (**C**) chlorophyll content, (**D**) proline content, (**E**) MDA content, and (**F**) relative conductivity in WT, UL, and *MbMYB4*-OE (S1, S3, and S5) *A. thaliana* under control condition (Cold 0 h) and cold stress (Cold 14 h). Asterisks above the error bars indicate an extremely significant difference between transgenic and WT plants (** *p* ≤ 0.01). The level of each index in WT was used as control.

**Figure 7 ijms-23-01794-f007:**
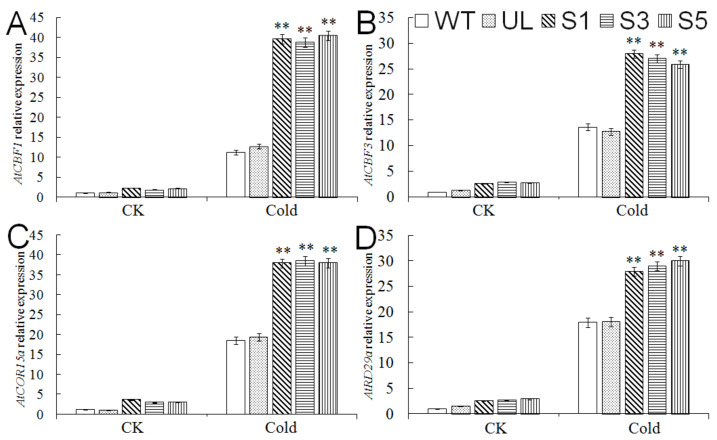
Expression levels of cold stress-related genes in WT, UL, and transgenic *A. thaliana* under cold stress. (**A**) Relative expression level of *AtCBF1,* (**B**) relative expression level of *AtCBF3*, (**C**) relative expression level of *AtCOR15a*, (**D**) relative expression level of *AtRD29a*. Data represent means of three replicates. Asterisks above columns indicate significant difference compared to WT (** *p* ≤ 0.01).

**Figure 8 ijms-23-01794-f008:**
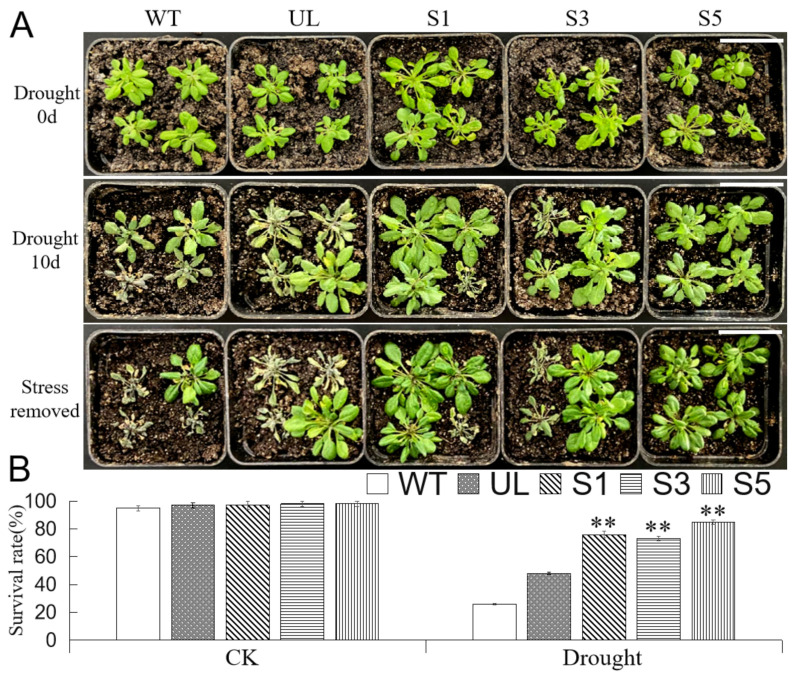
Overexpression of *MbMYB4* in *A. thaliana* improved drought tolerance. (**A**) Phenotypes of WT, UL, and *MbMYB4* transgenic *A. thaliana* lines (S1, S3, and S5) under control condition, drought stress, and stress-removed condition. Scale bar corresponds to 4 cm. (**B**) Survival rates of WT, UL, and transgenic *A. thaliana* under control condition and drought stress. Asterisks above columns indicate significant differences between transgenic *A. thaliana* (S1, S3, and S5), UL, and WT line (** *p* ≤ 0.01).

**Figure 9 ijms-23-01794-f009:**
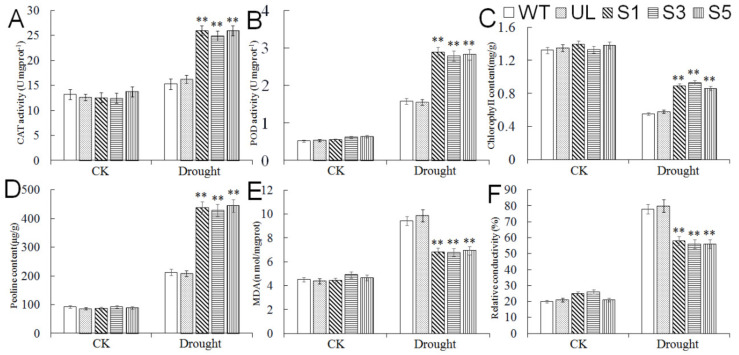
The levels of (**A**) CAT activity, (**B**) POD activity, (**C**) chlorophyll content, (**D**) proline content, (**E**) MDA content, and (**F**) relative conductivity in WT, UL, and *MbMYB4*-OE (S1, S3, and S5) *A. thaliana* under control condition (Drought 0 d) and drought stresses (Drought 10 d). Asterisks above the error bars indicate an extremely significant difference between transgenic and WT plants (** *p* ≤ 0.01). The level of each index in WT was used as control.

**Figure 10 ijms-23-01794-f010:**
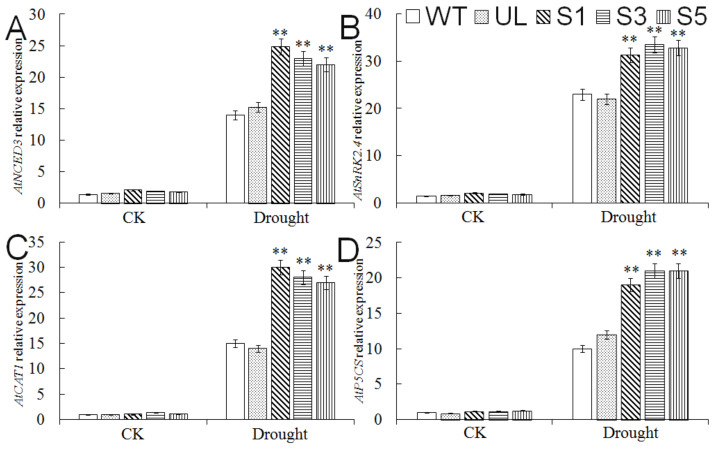
Expression levels of drought stress-related genes in WT, UL, and transgenic *A. thaliana* under drought stress. (**A**) Relative expression level of *AtNCED3,* (**B**) relative expression level of *AtSnRK2.4*, (**C**) relative expression level of *AtCAT1*, (**D**) relative expression level of *AtP5CS*. Data represent means of three replicates. Asterisks above columns indicate significant difference compared to WT (** *p* ≤ 0.01).

**Figure 11 ijms-23-01794-f011:**
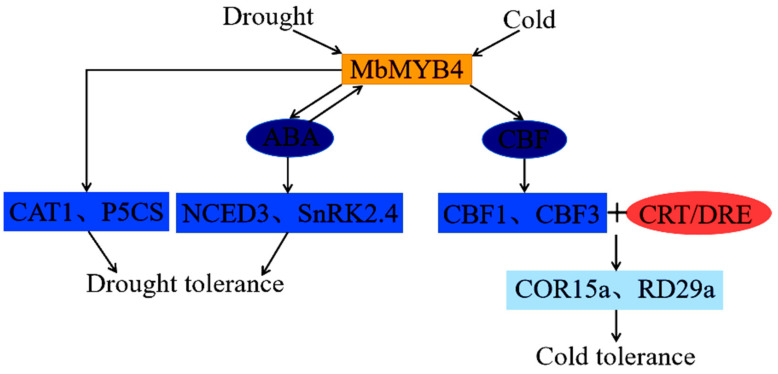
A possible model for the role of *MbMYB4* in cold and drought stresses. Drought stress induced the expression of *MbMYB4*, promoted the expression of ABA biosynthesis and signal transduction genes *NCED3* and *SnRK2.4*, and improved drought tolerance. At the same time, the production of ABA reversely induces the expression of *MbMYB4*. In addition, drought stress promotes the expression of key drought response genes *CAT1* and *P5CS*, and improves drought tolerance from another way. Cold stress induces the expression of *MbMYB4*, and *MbMYB4* binds to the promoter regions of *CBF1* and *CBF3* and promotes the binding of *CBF* gene to the CRT/DRE cis-acting elements of downstream genes, thereby activating the expression of downstream cold-responsive genes *COR15a* and *RD29a*, and enhances plant cold tolerance.

## Data Availability

Not applicable.

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
