# Peer review of "Overexpression of a *Malus baccata* MYB Transcription Factor Gene *MbMYB4* Increases Cold and Drought Tolerance in *Arabidopsis thaliana"

_ijms, 2022, doi:10.3390/ijms23031794_

Round 1

Reviewer 1 Report

The submitted Manuscript provides interesting results on how the heterologous expression of MbMYB4 increased tolerance to cold and drought for transgenic A. thaliana plants. The results are definitely worth to be published but the submitted Manuscript needs essential revisions and potentially several rounds of revisions.

1) Abstract.

Therefore, these results suggest that 30 MbMYB4 probably plays an important role in the response to cold and drought stress in A. thaliana 31 by enhancing the scavenging capability for reactive oxygen species (ROS).

MbMYB4 is not playing any role in A. thaliana since it is originally absent in Arabidopsis. Please, rephrase adding word “transgenic” at least.

2) Please, provide more references with 2-3 references for each statement. The Reviewer found the one

The MYB Transcription Factor Superfamily of Arabidopsis: Expression Analysis and Phylogenetic Comparison with the Rice MYB Family https://link.springer.com/article/10.1007/s11103-005-2910-y

which makes comparisons between rice and Arabidopsis. The same idea would be logically interesting for the comparison of MYB transcription factors from Malus and from Arabidopsis. So, the Reviewer suggests to add a figure with MbMYB4 and groups of TFs from Arabidopsis to shew where MbMYB4 lies and positioned.

3) The main question is why and how the TF MbMYB4 influences the network of TFs in Arabidopsis. Please, provide more discussion and reasons why the heterologous expression of MbMYB4 increased and not decreased the tolerance for Arabidopsis.

4) Please, add the summarizing scheme and graphical abstract for the submitted MS.

5) Figure 2. Please, provide-make the better quality figure for the suggested 3D structure of MbMYB4, it would be good to mark/indicate the corresponding domains from figure 2B at the 3D structure. Pls, compare to the good figures from the corresponding good journals (based on the present impact factor). The present quality of the whole figure is too low.

6) Figure 3. Please, provide an extra figure with many cells of the cell layer depicted, the Reviewer would like to see the efficiency of transformation and reproducibility of the data.

7) Figure 4. Please, provide (potentially as a supplementary material) the figure of the Malus plantlets.

8) Seven days after the stress was deleted (recover), most of 162 WT and UL died, the survival rates were only 35.83% and 26.17%, while the average sur-163 vival rate of transgenic A. thaliana was 76.77% (Figure 5C).

The mentioned % are not correct, the two digital points after a dot would make sense when over 10,000 plants had been studied. So far it 26% and 77% based on the figure 5.

9) Figure 5. 5B, it is not a recovery but the same dying plants. Please, rename somehow to “stress removed” or so.

10) 2.6. Overexpression of MbMYB4 in A. thaliana improved to Drought Stress Tolerance

Improved drought tolerance.

11) The same as 10 for 2.4.

12) Please, provide more description for the methods how the stresses were applied.

low temperature stress (4℃376 ), salt stress (200 mM NaCl), dehydration stress (20% PEG), high temperature stress (37℃377 ) and ABA stress (50 μM).

13) Pls, provide more description in the methodical part for all the sections. Potentially the part could be added as supplementary ones.

14) Please, rewrite the methodical part in the past tense excluding imperatives.

15) Figure S1. Several genes from Arabidopsis are to be added for comparisons and pointing to the conservative domains.

16) Potentially the submitted MS could be considered further after the suggested revisions. It requires revisions from the very beginning of the introduction.

Reviewer 2 Report

In this manuscript authors studied that overexpression of a Malus baccata MYB Transcription Factor Gene MbMYB4 Increases Cold and Drought Tolerance in Arabidopsis thaliana. In this study, Malus baccata (L.) Borkh was used as the research material, and a gene MbMYB4 of the MYB family was cloned from it. The MbMYB4 gene encoded a protein of 253 amino acid residues with a theoretical isoelectric point of 6.84 and a predicted molecular mass of 28.843 kDa. Subcellular localization showed that MbMYB4 was localized in the nucleus. In addition, the use of quantitative real-time PCR technology found that the expression of MbMYB4 was enriched in young leaf and root, and was highly affected by cold and drought treatments in M. baccata seedlings. When MbMYB4 was introduced into Arabidopsis thaliana, it greatly increased a transgenic plant's low temperature and drought tolerance. Under low temperature and drought stresses, the proline and chlorophyll content, peroxidase (POD) and catalase (CAT) activities of transgenic A. thaliana increased significantly, the content of malondialdehyde (MDA) and the relative conductivity decreased significantly, indicating that the plasma membrane of transgenic A. thaliana was less damaged. Therefore, these results suggest that MbMYB4 probably plays an important role in response to cold and drought stress in A. thaliana by enhancing the scavenging capability for reactive oxygen species (ROS). I have a few comments and concerns for the betterment of this manuscript.

  1. Why do authors need to make transgenic plants in Arabidopsis but not in the Malus to better functional charecterization of this gene?
  2. In figure 1, Font A is inserted in-between figure. Please correct it.
  3. In figure 4, the control has some expression; however, in figure 5A, Control has 0 expressions; how it is possible also CaMV 35S promoter is very strong how the author is going to explain the only 1.5 fold induced expression in Ox lines. Also, How did the authors choose 3 Ox lines (Copy number )? Which generation of seeds do authors use for the experiments?
  4. What is unloaded (UT) lines? The authors should have used an empty vector control line along with WT.
  5. Make one hypothetical figure that includes a finding of this study.

Change at L18 from

affect their growth to affects their growth.

L31 important role in the response to cold to important role in response to cold.

L33, L167, L323, L376 low temperature to low-temperature.

In this manuscript I found Plagiarism at L21-22, L30-32, L42-43, L45-46, L80-81, L84-85, L97-99, L121-125, L130-131, L133-136, L139-141, L143-146, L148-149, L179-180, L181-184, L188-190, L404-405, L441-442, L443-446,

Round 2

Reviewer 1 Report

The Authors reasonably answered to the series of the posed questions but, unfortunately, forgot to include the promised Figure 11, Figure S2, figure 1C etc, so the MS requires an extra round or extra several rounds of revisions.

More:

1) In the added part. The open reading frame

23 (ORF) of MbMYB4 is 762 bp, encoding 253 amino acids; sequence alignment results

ORF of 762 bp should code 254 aa, 762/3 = 254. Pls, explain the reasons.

2) Placed the

332 cultured onion cells under a confocal microscope and observed the fluorescence signals of the control

333 protein and MbMYB4-GFP fusion protein, so as to determine the location of these proteins in the cell.

It’s not readable.

The Reviewer is not able to evaluate the MS without the added material and extra checks by the Authors. The next round will be the recommended rejection from the Reviewer.

Reviewer 2 Report

I am happy with the author's reply and the manuscript improved a lot and can be accepted in its current format. The updated manuscript does not contain any figures and its format is different.

Round 3

Reviewer 1 Report

The Authors comprehensively responded to the questions and comments, so the present Reviewer is glad to endorse the submitted Manuscript for publication.

The Authors could improve the quality of Figures 2 and 11, add more complete description to figure legend for summarizing figure 11 and add the linear scale bar for additional figure S2, it would improve the appreciation of the Manuscript by the potential Readers and attract more attention to the good presented results.
